# Visualizing nanometric structures with sub-millimeter waves

Alonso Ingar Romero[1], Amlan kusum Mukherjee [1✉], Anuar Fernandez Olvera[1], Mario Méndez Aller[1] & Sascha Preu [1✉]

The resolution along the propagation direction of far field imagers can be much smaller than the wavelength by exploiting coherent interference phenomena. We demonstrate a height profile precision as low as 31 nm using wavelengths between 0.375 mm and 0.5 mm (corresponding to 0.6 THz–0.8 THz) by evaluating the Fabry-Pérot oscillations within surface-structured samples. We prove the extreme precision by visualizing structures with a height of only 49 nm, corresponding to 1:7500 to 1:10000 vacuum wavelengths, a height difference usually only accessible to near field measurement techniques at this wavelength range. At the same time, the approach can determine thicknesses in the centimeter range, surpassing the dynamic range of any near field measurement system by orders of magnitude. The measurement technique combined with a Hilbert-transform approach yields the (optical) thickness extracted from the relative phase without any extraordinary wavelength stabilization.

[1] Department of Electrical Engineering and Information Technology, Technical University of Darmstadt, 64283 Darmstadt, Germany.
✉email: amlan.mukherjee@tu-darmstadt.de; preu@imp.tu-darmstadt.de

Optical surface topography plays a key role in many fields, ranging from large distance, airborne, or satellite imaging and monitoring of earth[1,2] down to table-top, contactless and non-destructive characterization of objects such as the roughness and shape of surfaces[3,4] and inspection and quality control[5]. For the latter, optical far-field methods underwent a steady evolution and are amongst the main methods of choice, in particular, when high aspect ratio combined with nanometric depth resolution is required[6,7]. In the surface plane, perpendicular to the propagation direction, the resolution of far-field imagers is limited by the Abbe limit or Rayleigh criterion[8,9] leading to a spatial resolution that is larger than the imaging wavelength. Along the propagation direction, however, exploiting the interference properties of light enables resolution of dimensions much smaller than the wavelength. When two waves interfere they create an interference pattern, where one interference fringe refers to a path length difference of one wavelength between the two waves. By trimming the path length difference to follow the interference maximum, the resolution can be a small fraction of the wavelength. It is then rather limited by how fine and how accurate the path length difference between the two waves can be trimmed or measured. Typical system architectures are, e.g., Michelson Interferometers where the object is placed in one of the interferometer arms or replaces a mirror in one arm while the other arm length is swept. Common examples of such setups are white light interferometers[10] and optical coherence tomography[11–13] (OCT) with a depth resolution in the range of 0.1–1 μm using visible or broadband light, i.e., still of the order of the wavelength. In the Terahertz (100 GHz–10 THz) range, where the research reported in this manuscript has been performed, the best OCT resolution demonstrated so far is much worse, namely 43.9–220 μm[14,15] attributed to the about a factor of 1000 longer wavelength. Using monochromatic light sources allows for surpassing the wavelength limit in terms of depth resolution. Frequency-modulated continuous-wave (FMCW) techniques allow for resolving an optical thickness of $nd = \frac{c_0}{2B}$ [16], $c_0$ being the speed of light in vacuum and $B$ the measurement bandwidth. But even with a bandwidth of a several THz, the optical thickness resolution lies above 10 μm. Still, the method can be highly accurate, with a reported measurement error as low as 360 nm[17]. Using a heterodyne technique with advanced data processing, the measurement error of a single frequency system running at 600 GHz was reduced to about 0.5 μm[18]. However, single frequency systems are prone to $2\pi$ errors when surfaces with thickness variations larger than half the wavelength are examined. After decades of evolution, visible-light phase-shift interferometers and swept-frequency interferometers yet achieve a resolution in the range of 0.1 nm[6], corresponding to $\sim\lambda/5000$. With advanced techniques such as locking to high-quality-factor external cavities (Fabry–Pérot Interferometers[19]), by using multiple roundtrips[20] or simply by using much shorter wavelengths, e.g., X-rays[21], (sub-)picometer resolution has been achieved, however, requiring a confinement of the measurement range to a few μm. Terahertz far-field systems so far achieved a thickness resolution around 0.2–10 μm[22–25] typically exploiting a bandwidth beyond 3 THz, yet with standard deviations as low as 40 nm[26]. Only Terahertz near-field methods achieve few nanometer resolution[27,28]. In this paper, we demonstrate a Terahertz far-field system working at imaging wavelengths between 0.375 mm and 0.5 mm with the ability to visualize surface structures with heights of only 49 nm ($\sim\lambda/10000$). The root mean square error (i.e., the precision) is as small as 31 nm ($\sim\lambda/15000$) on a silicon sample that is as thick as 0.5 mm, despite the employed bandwidth of 0.2 THz is at least 15 times smaller than that of the THz systems with the best thickness resolution to date.

## Results and discussion

**Principle of operation.** The presented surface topography technique uses a continuous-wave Terahertz homodyne photomixing spectrometer (TOPTICA Photonics TeraScan[29]) as illustrated in Fig. 1a. It consists of a photomixer emitter (WIN-PD) and a photoconductive receiver (either a commercial photoconductor from Toptica Photonics/Fraunhofer Heinrich Hertz Institute or an ErAs:InGaAs photoconductive receiver[30,31]). Both photomixer source and photoconductive receiver are driven by a pair of 1550 nm lasers that differ in frequency by the desired Terahertz frequency $f_{THz}$. The frequency is tunable simply by tuning one or both laser frequencies. The photomixer source converts the beat note of the two lasers to a Terahertz beam with the same frequency $f_{THz}$[32]. The Terahertz beam propagates through the experimental setup, consisting of a parabolic mirror that collimates the emitted beam, a first TPX lens that creates a focal point where the sample is mounted, a second TPX lens that re-collimates the beam, and a

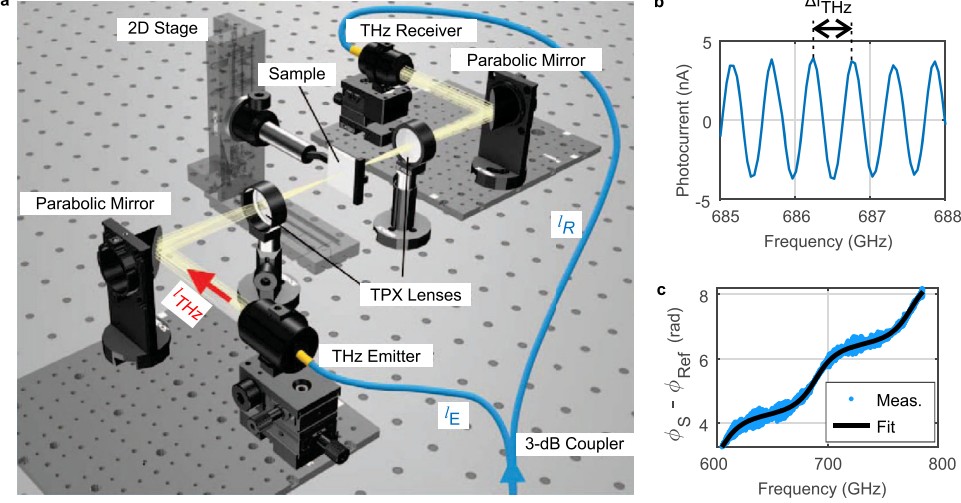

**Fig. 1 Homodyne fringes in the measurement setup and exemplary data-fitting. a** Homodyne photomixing setup. The lengths $l_s = n_F l_E + l_{THz}$ and $l_r = n_F l_R$ correspond to the optical path lengths of the two Mach–Zehnder interferometer arms, $n_F$ being the refractive index of the optical fibers after the 3 dB coupler. **b** Raw homodyne data. **c** Extracted phase of a sample with a thickness of $d_S = 509.3$ μm and a refractive index of $n_S = 3.416$ (silicon). The reference measurement of the empty setup has already been subtracted.

final parabolic mirror that focuses it onto the receiver. The setup is operated in air without any dry nitrogen flooding. The receiving photoconductor is attached to an antenna that converts the received Terahertz field $E_{THz}$ into a DC current proportional to the field amplitude[30]. The photoconductor is essentially a mixer that multiplies the received Terahertz field with the envelope of the lasers' beat signal. Source and receiver are therefore phase-locked, as the same pair of lasers is used to operate them. The whole setup can be considered as a Mach–Zehnder or a phase-shift Interferometer. The DC component of the receiver reads $I_r(f_{THz}) \sim E_{THz}(f_{THz})\cos(\triangle\vartheta(f_{THz}))$, with the phase difference of the two interferometer arms $\triangle\vartheta(f_{THz}) = \frac{2\pi}{c_0}f_{THz}\triangle(nd)$. The phase difference is caused by the optical path length difference $\triangle(nd) = l_s - l_r$ as illustrated in Fig. 1a. An exemplary measurement is illustrated in Fig. 1b. For non-zero optical path length difference, $\triangle(nd) \neq 0$, the phase of the received current oscillates with frequency. By sweeping over at least one oscillation cycle, Terahertz amplitude, phase and oscillation period can be recovered without any mechanical delay line, which is otherwise imperative to obtain phase information. Thickness variations of a sample under test introduced in the Terahertz path will modify the optical path length difference $\triangle(nd)$ and thus cause a change in the phase and oscillation period. We exploit this fact to measure the height of surface structures by recording the phase of the interferogram. A robust way to acquire the phase information is the Hilbert transform of the received current versus frequency. Simply speaking, the Hilbert transform $\mathscr{H}$ transforms an oscillating signal like $I_r(f_{THz})$ into a complex signal, $I_r(f_{THz}) \rightarrow A(f_{THz})\exp i\triangle\vartheta(f_{THz})$, where the amplitude of the oscillation is $A(f_{THz})$ and the argument of the complex signal is the required phase $\triangle\vartheta(f_{THz})$. Interestingly, phase changes can be resolved at a fraction of the oscillation period[33], enabling extremely high resolution. Figure 1c illustrates the extracted phase from the data in Fig. 1b, corresponding to a sample with plane-plane surfaces, a thickness $d_S = 509.3\,\mu m$ and a refractive index $n_S = 3.416$ (silicon wafer, optical thickness $n_s d_s = 1.740$ mm). The phase of a reference measurement with an empty setup has been subtracted from the sample measurement for better visibility. The phase shows a linear increase with a slope of $\frac{\partial\triangle\vartheta}{\partial f_{THz}} = \frac{2\pi}{c_0}(n_s - 1)d_s = 25.79$/THz and an oscillation around it. The linear part corresponds to a path length increase by $(n_s - 1)d_s$ due to the presence of the sample in the setup instead of air. The thickness error obtained with a fit to the linear part is proportional to the total optical path length difference $\triangle(nd)$ in the interferometer when $n_s d_s \ll \triangle(nd)$, (c.f. Eq. (13)). The oscillation around the linear slope is caused by Fabry–Pérot interference within the plane-plane sample. Fitting the Fabry–Pérot oscillations with the theoretical expression shown in Eq. (10) allows extracting both

optical thickness (from the periodicity) and refractive index (from the finesse). The geometric thickness is simply their ratio. In an ideal scenario, the error of the optical thickness of the sample $n_S d_S$ is dominated by how precise the free spectral range can be measured, not by $\Delta(nd)$. Under typical measurement conditions, $n_S d_S$ is several orders of magnitude smaller than $\Delta(nd)$, therefore offering much better thickness resolution.

**Hilbert-transform-based imaging**. As there are no photoconductive cameras available in the Terahertz range, we measure in a single pixel configuration, scanning the sample through the focal spot between the two TPX lenses in two lateral dimensions. We limit the measurement frequency interval of the swept source interferometer to 0.6–0.8 THz for the sake of measurement time, i.e., using a small fraction of the frequency range usually employed in thickness measurements. Figure 2a shows the optical thickness image of a 10.7 ± 0.2 μm tall Siemens star (measured mechanically with a Dektak profilometer) etched into the surface of a high-resistivity (HR) silicon wafer (literature value of $n_{obj} = 3.416$). The Terahertz image was obtained by fitting the Fabry–Pérot interference pattern (Eq. (10)), where the refractive index and the optical thickness are the fitting variables. The thickness of the silicon host wafer is not constant, its optical (physical) thickness varies by 14 μm (4.1 μm) across the measured sample area. Averaging over the thickness variation, we obtain a host wafer optical thickness of 1.7866 ± 0.007 mm, confirming the mechanically determined geometric thickness of 520 ± 5 μm (which corresponds to an optical thickness 1.7763 ± 0.017 mm using the text book refractive index of silicon of 3.416) i.e., an accuracy better than 0.6%. Figure 2b shows the measured refractive index image of the sample. As the finesse is affected by the structure's edges, a *simultaneous extraction* of refractive index and geometrical thickness does not yield precise results there, but the edges of the structure become clearly visible, enabling to use this methodology for edge detection. The refractive index determined from the unetched areas of the host wafer is 3.47 ± 2%, in good agreement with the aforementioned literature value. We then use this refractive index as a fixed value to determine a geometrical thickness of 10.66 ± 0.3 μm for the etched Siemens star from its fitted optical thickness of 37.0 ± 1 μm, as depicted in Fig. 2c. This value is in excellent agreement with the profilometer measurements. We note that simultaneous determination of geometrical thickness and refractive index of the etched structure is also possible but yields a higher thickness error of 10.5 μm. The lateral resolution is still diffraction-limited as shown by the blurring and reduced depth in the center of the Siemens star. This accuracy was obtained with a 30 dB signal-to-noise ratio per measured pixel and a path length difference $\triangle(nd) = 55$ cm.

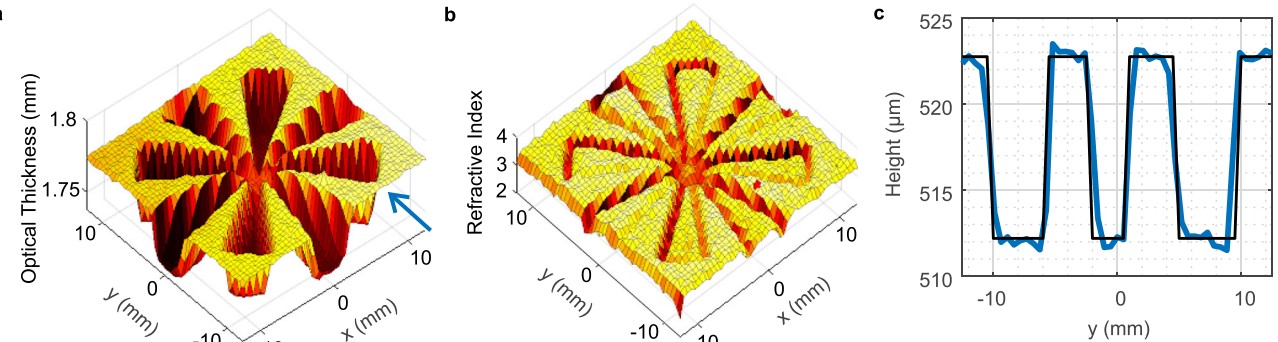

**Fig. 2 Measurement of a 10 μm etched Siemens star on silicon wafer. a** 3D Terahertz image of a Siemens star with a depth of 10.7 μm etched into a 520 μm thick silicon wafer. The vertical axis shows the optical thickness. **b** 3D image of the refractive index of the silicon Siemens star. **c** Extracted physical thickness of the silicon Siemens star for a cut along the line x = 8 mm indicated by the blue arrow in (**a**).

The structure height precision ($\Delta h$) is evaluated by Eq. (13) to be proportional to the optical path length difference $\Delta(nd)$ and the ratio of frequency instability/drift of the DFB lasers ($\delta f_{THz}$) to the measurement bandwidth ($f_{scan}$), $\Delta h = \frac{\Delta(nd)}{n_{obj}} \cdot \frac{\delta f_{THz}}{f_{scan}}$. Hence, for the same bandwidth, the measurement precision can be significantly improved by using a smaller path length difference. We therefore image a 240 thick SiN layer with reduced $\Delta(nd) = 7.25$ cm with a roughness of the order of 20 nm on a $509 \pm 5$ μm thick HR silicon wafer (optical thickness 1.739 mm), and a $49 \pm 1$ nm thick SiC layer (refractive index $n_{obj} \sim 3.1 \pm 0.4$[34], optical thickness $\sim 151.9 \pm 20$ nm) on a $525 \pm 5$ μm thick HR silicon wafer with $\Delta(nd) = 2.4$ cm. Both object thicknesses were determined with a Dektak profilometer. Both layers were grown using chemical vapor deposition (CVD). Figure 3a shows that the substrate is warped with a thickness variation in the range of 5 μm. The 240 nm thick Siemens star is hidden within the warped background. Subtracting the warped background (shown in Fig. 3e) reveals the 240 nm thick Siemens star (Fig. 3b) with an optical thickness of $490 \pm 160$ nm (Fig. 3c), corresponding to a refractive index of $2.04 \pm 0.19$ for the physical thickness determined with the Dektak profilometer. The refractive index agrees very well with the near-infrared value for SiN but it is much less than the terahertz value reported in the literature ($n_{obj} \sim 2.75$[35]). The lower value may be due to CVD process used for growth that provides less dense films. Figure 3d shows an optical micrograph of the Siemens star. A further reduction of the optical path length difference by a factor of three ($\Delta(nd) = 2.4$ cm) enables to visualize the 49 nm thick Siemens star of CVD-grown SiC on silicon (Fig. 3f, g). The star is hardly visible with a geometrical thickness precision of 31 nm, already showing the ultimate limits of the approach. Figure 3h depicts the optical micrograph, also revealing some roughness of the very thin structure.

Finally, we evaluate the applicability of the approach to a thicker, unstructured sample with lower refractive index, namely Teflon, with a thickness of $10.7 \pm 0.15$ mm, determined with a caliper. The thickness error is due to a non-perfect planarity of the sample. We now only use the *slope* of the recorded phase. In order to extract the physical thickness from the slope, previous knowledge of the refractive index is required. Using the literature value of 1.44 for the refractive index[36] results in a thickness of $10.84$ mm $\pm 1.1$ μm (at $\Delta(nd) = 55$ cm) at the sample center, within the error of the caliper measurement. A Terahertz time domain spectroscopy measurement provided a refractive index of $n_{obj} = 1.4336 \pm 0.0015$ with a thickness of $10.72 \pm 0.05$ mm. We therefore conclude that the accuracy is in the range of 1% for both the refractive index and the thickness.

**Robustness**. The implemented methodology circumvents several flaws of any unstabilized photomixer setup, which is, in the end, the reason for the extremely high thickness resolution. The frequency values read from the Toptica system are slightly inaccurate as their values originate from a look-up-table that maps a set laser temperature to a corresponding THz frequency $f_{THz}$, which were calibrated at a low tuning speed of the set temperature. At high tuning speeds, as the ones used in this experiment, the actual frequency lags behind the set frequency. The slope is immune against offsets as well as the evaluated periodicity caused by Fabry–Pérot oscillations as long as the system shows high repeatability. For the latter, however, allowing for a small frequency offset may improve the fitting accuracy. Random imperfections of the frequency or amplitude information of the $N = 4000$ frequency points scanned over a 0.2 THz frequency window have only very minor impact on the data quality as all $N$ data points of each frequency sweep are fitted at once, reducing the error roughly by $1/\sqrt{N}$. Laser drifts on time scales much longer than the measurement duration of 24 s, such as the ones caused by temperature changes in the laboratory, will not affect the thickness accuracy as each trace is fitted individually. The distributed feedback lasers are only thermally stabilized with PID controllers to about $209 \pm 50$ kHz during the acquisition time of 24 s. Such drifts do reduce the precision. The dependence on the laser stability is remarkable, as employing stabler laser systems, such as continuous-wave frequency combs[37] will enable orders of

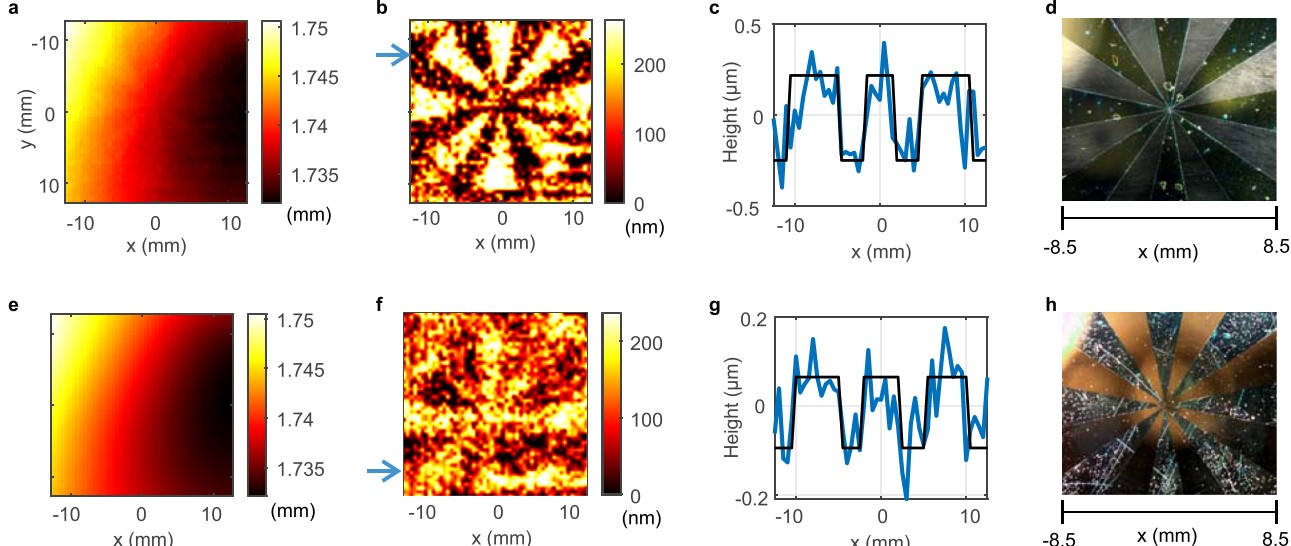

**Fig. 3 Measurements of nanometric SiN and SiC Simens stars deposited on a silicon wafer. a** Recorded Terahertz image. **b** Terahertz image of the 240 nm thick SiN Siemens star after subtracting the warping of the silicon wafer (shown in **e**) and image enhancement. **c** Actual optical height profile of the line indicated by the blue arrow in (**b**). **d** Optical micrograph of the center of the 240 nm thick SiN Siemens star on the silicon wafer. **e** Warping of the silicon wafer. **f** Terahertz image of the 49 nm thick SiC Siemens star on a silicon wafer. **g** Actual optical height profile of the line indicated by the blue arrow in (**f**), after subtraction of warped background. **h** Optical micrograph of the 49 nm thick SiC Siemens star. The color scales of the terahertz images represent the optical thicknesses.

magnitude higher precision than reported here. A large impact on data quality is due to undesired standing waves in the Terahertz setup. The standing waves cause fluctuations in the frequency domain that are nicely visible in the blue oscillations around the red fit shown in Fig. 1c. The optical path lengths of such oscillations are much larger than the sample thickness, with an oscillation period typically 1–2 orders of magnitude shorter than the Fabry–Pérot oscillations. Since the fitting algorithms used for both the slope and the Fabry–Pérot oscillations are applied to the whole frequency spectrum, the averaging effect strongly reduces the deteriorating influence of standing waves, as it is evident from the red line in Fig. 1c. The appearance of undesired reflections can partly be compensated by subtracting a reference measurement recorded before the first pixel or after the final pixel, respectively. We remark that a certain refractive index contrast is required for applying the Fabry–Pérot evaluation method. The slope fit can always be applied and also works for other measurement geometries such as in reflection mode of highly reflective or absorptive samples where no Fabry–Pérot interference occurs. In principle, the optical thickness can be determined without a priori knowledge, $n_{obj}\triangle h$ as long as at least one Fabry–Pérot fringe falls within the measurement range. The disentanglement of thickness and refractive index can be achieved by a measurement of the refractive index where no structure is apparent. Its relative error, $\triangle n_{obj}/n_{obj}$, translates to a constant, systematic relative thickness error of the surface structure. More accurate values require a priori knowledge of the substrate's refractive index.

Measurement precision can be further improved by scanning over a larger bandwidth. As shown in Eqs. (13) and (14), the measurement precision $\triangle h_{FP}$ is inversely proportional to the scanning bandwidth ($f_{scan}$). Additionally, scanning over a larger frequency range will increase the fitting efficiency ($\eta_{fit}$) as higher numbers of Fabry–Pérot fringes will ensure a better fit. Thus, by simply increasing the scanning bandwidth to 2–4 THz, which is usually the case for the state-of-the-art thickness measurement systems, the measurement precision can be improved to sub-nanometer levels, given that other measurement imperfections do not limit the precision. However, such bandwidths are usually only available in pulsed Terahertz time domain spectroscopy (TDS) systems. A measured standard deviation of 40 nm was recently reported by Molter et al.[26] employing such a system. Although Liebermeister et al. have recently reported a CW system with a bandwidth 4 THz[24], using such huge bandwidth will also immensely increase the overall scan duration for the proposed single-pixel based imaging system. Last but not least, no multi-layer structures were characterized. Analyzing these will require larger bandwidth, genetic algorithms assisted multilayer modeling[38], and potentially some a priori knowledge of the structures[39].

In conclusion, the presented Hilbert-transform-based homodyne far-field imaging system enables two measurement modes. A coarse measurement mode evaluates the slope of the Hilbert-transform-enabled phase profile with an excellent geometrical thickness resolution in the 1 μm range, corresponding to about 1/400 vacuum wavelengths, despite severe undesired standing waves in the setup. This resolution is competitive with any other far-field method in the Terahertz domain. The resolution can be further enhanced by using smaller path length differences in the Mach–Zehnder interferometer arms without any effects on the Hilbert-transform method, which is able to extract phases at a fraction of a single interference fringe. This methodology can also be applied for surface topology in reflection mode, even for highly reflecting or highly absorbing materials. A finer measurement mode exploits Fabry–Pérot cavity-enhanced detection to improve

the resolution by about 1.5 orders of magnitude, with a demonstrated rms error of only 31 nm, verified by visualizing structures as small as 49 nm on a 525 μm silicon substrate at Terahertz frequencies. This corresponds to an aspect ratio of >10,000 and a height resolution better than 1/7500 vacuum wavelengths. Thickness measurement of a 1 cm thick plastic sample points out the extreme height dynamic range of more than a factor of 200,000. Even thicker samples are measureable. The used setup is very simple and robust, requiring just a standard homodyne photomixing setup. The method can also be transferred to coherent electronic systems, such as vector network analyzers and even to frequency swept interferometers in the visible or other spectral domains. Advanced data extraction techniques, such as genetic algorithms assisted multilayer modeling[38] or neural networks that are yet implemented in many state-of-the-art Terahertz thickness measurement systems may further improve the precision.

## Methods

**The homodyne photomixing setup.** Two lasers of equal polarization and ideally equal power $P_0$ with frequencies $f_1$ and $f_2 = f_1 + f_{THz}$ are superimposed. This leads to a laser beat note[32] of

$$P_L(t) = P_0[1 + \cos(2\pi f_{THz} t + \vartheta)] \tag{1}$$

where $\vartheta$ is the relative phase between the two laser signals. The pin-diode-based source absorbs the laser power, generating a DC current as well as an AC current at the beat frequency,

$$I_{AC}(t) \sim P_0\cos(2\pi f_{THz} t + \vartheta_S). \tag{2}$$

The index "s" at $\vartheta_S$ indicates the phase of the photocurrent at the source, that is subsequently radiated by an antenna in the form of a Terahertz field, $E_{THz}(t) \sim I_{AC}(t)$, transmitted to the receiver, and there converted to an AC bias. During propagation, an additional phase of $\vartheta_p = \frac{2\pi f_{THz}}{c_0}\sum_i n_i d_i$ is acquired, where $n_i$ and $d_i$ are the refractive indices and thicknesses of all structures along the propagation to the receiver, and $c_0$ the speed of light in air. In Fig. 1a, $\sum_i n_i d_i = l_{THz}$. The laser beat note operating the receiver essentially works as phase-locked local oscillator (LO) that modulates its conductivity as $\sigma_{rec}(t) \sim P_L(t) = P_0\left[1 + \cos(2\pi f_{THz} t + \vartheta_R)\right]$, where the receiver phase $\vartheta_R$ may differ from that of the source. The conductivity modulation combined with the Terahertz bias results in a drift current $I_R(t) \sim E_{THz}(t)\sigma_{rec}(t)$, effectively causing a multiplication of the laser LO and the received Terahertz signal. We note that this mixing process differs from that occurring in nonlinear media or Schottky diodes as it only generates a DC component and two remaining Terahertz components and no further higher order mixing components. The DC receiver component,

$$I_{R_{DC}} \sim E_{THz}^0 \cos(\triangle\vartheta), \tag{3}$$

is proportional to the product of received Terahertz field magnitude $E_{THz}^0$ and its phase $\triangle\vartheta$. The phase being equal to

$$\triangle\vartheta = \vartheta_S + \vartheta_P - \vartheta_R = 2\pi f_{THz}\left[n_F(l_E - l_R) + \Sigma_i n_i d_i\right]/c_0 = 2\pi f_{THz}\triangle(nd)/c_0, \tag{4}$$

where $c_0$ is the vacuum speed of light, $n_F$ is the refractive index of the fiber of lengths $l_E$ and $l_R$ connecting the source and the receiver measured from the 3 dB splitter, respectively, and $\triangle(nd)$ is the total optical path length difference between the source- and receiver interferometer arm. For $\triangle(nd) \neq 0$ the received current oscillates with terahertz frequency as $I_R \sim E_{THz}^0 \cos(2\pi f_{THz}\triangle(nd)/c_0)$ as shown in Fig. 1b. The frequency spacing between two fringe maxima, $\triangle f_{THz}$, allows to extract the total path length difference $\triangle(nd) = c/\triangle f_{THz}$. The fringe period can easily be set by the fiber lengths connecting source and receivers or by the length of the Terahertz path. In the current measurement, the fringe period was varied between 0.545 and 12.5 GHz, corresponding to an optical path length difference between 55 cm and 2.4 cm. Recording the period of the oscillations rather than a phase at a constant Terahertz frequency removes the usual $2\pi$ uncertainty associated with single frequency measurements. A key point for thickness measurement is that the Terahertz path length, the fiber lengths and the imaging optics remain constant throughout all measurements. Only sample thickness variations cause changes in $\triangle\vartheta$. For offset removal, we record $\frac{\partial\triangle\vartheta}{\partial f_{THz}} = 2\pi\triangle(nd)/c_0$, rather than $\triangle\vartheta$ and perform a nonlinear-least squares fit to the data. A key challenge, however, is a solid, accurate method to extract the phase from the measured fringe pattern.

**The Hilbert transform.** The Hilbert transform is very accurate and offers extreme resolution when used to extract the phase of a periodic signal[33]. The achievable resolution is well below one fringe period, and that fact is exploited here. The Hilbert-Transform $\mathscr{H}\{x(t)\}$ of a real-valued signal $x(t)$ is defined as the

convolution of $x(t)$ with the function $h_{\mathscr{H}}(t) = \frac{1}{\pi t}$:

$$\mathscr{H}\{x(t)\} = x(t) * \frac{1}{\pi t} = \frac{1}{\pi}\int_{-\infty}^{\infty}\frac{x(\tau)}{t-\tau}d\tau \qquad (5)$$

In frequency domain, this leads to the following relation:

$$\mathscr{F}\{\mathscr{H}\{x(t)\}\} = \mathscr{X}(i\omega) \cdot \mathscr{F}\left\{\frac{1}{\pi t}\right\} = -i\,\mathrm{sgn}(\omega) \cdot \mathscr{X}(i\omega), \qquad (6)$$

where $\mathscr{F}\{\bullet\}$ is the Fourier-transform operator. Using the obtained result as the imaginary part and $\mathscr{X}(i\omega)$ as the real part gives a complex-valued signal whose spectrum is zero at negative frequencies, i.e., the analytic form of $\mathscr{X}(i\omega)$. Following the linearity of the Fourier transform, the analytic form of $x(t)$ can be calculated from its Hilbert transform as

$$z(t) = x(t) + i\mathscr{H}\{x(t)\}. \qquad (7)$$

Extracting the argument of this complex-valued signal and applying a phase unwrapping algorithm to avoid artificial jumps of $2\pi$ yields the instantaneous phase information of the input signal $x(t)$.

**Transmission phase of a Fabry–Pérot resonator**. The investigated plane-plane samples are de facto Fabry–Pérot (FP) cavities, though with a fairly low quality factor and low finesse ($F = 2.3$). The (complex-valued) field transmission coefficient of a plane wave with wavenumber $k = k_0 n_s$ through such cavity is

$$t_{tot} = \frac{t_1 t_2 \exp(ik_0 n_s d_s - \alpha d_s)}{1 - r_1 r_2 \exp(2ik_0 n_s d_s - 2\alpha d_s)}, \qquad (8)$$

where $t_{1,2}$ (and $r_{1,2}$) are the field transmission (reflection) coefficients for the first and second interface, respectively, $\alpha = 2k_0\kappa$ is the absorption coefficient of the material's cavity with complex refractive index $\bar{n}_s = n_s + i\kappa$ and total thickness $d_s$. Both the amplitude information as well as the phase information can be used to evaluate the optical thickness of the sample, which includes the substrate in the case of the surface-structured high resistivity silicon wafers. The phase of the transmitted signal is $\varphi = \arg(t_{tot})$. In order to demonstrate the working principle of the concept, we use a FP cavity formed by a material with a real-valued refractive index $n$ in air with negligible material loss, thus $|r_1 r_2| = R = \left(\frac{n-1}{n+1}\right)^2$, $t_1 t_2 = T = 1 - R$ and $\alpha = 0$. The phase caused by the FP cavity then becomes

$$\varphi = k_0 n_s d_s + \arctan\frac{R\sin(2k_0 n_s d_s)}{1 - R\cos(2k_0 n_s d_s)}. \qquad (9)$$

The recorded phase shows an oscillation around a linear increase as shown in Fig. 1c, where the free spectral range is $FSR = c_0/(2n_s d_s)$. The factor of two originates from the cavity round trip. Taking into account that we measure the relative phase between the two interferometer arms, the recorded phase difference becomes

$$\Delta\vartheta_{FP}(f_{THz}) = \arctan\frac{R\sin(2mf_{THz})}{1 - R\cos(2mf_{THz})} + (m + m_0)f_{THz} + const, \qquad (10)$$

where the first part is the phase modulation due to the Fabry–Pérot resonator and the other terms are due to the path length difference in both interferometer arms, $\Delta(nd)$, the sample thickness and an offset caused by phase unwrapping that does not start at 0 Hz. The parameter $m = \frac{\partial \Delta\vartheta}{\partial f_{THz}} = \frac{2\pi n_s d_s}{c_0}$ represents the slope of the phase caused by the sample whereas $m_0 = \frac{2\pi[\Delta(nd) - n_s d_s]}{c_0}$ collects all phase contributions other than that from the sample, mostly originating from $\Delta(nd)$. A fit of the first term delivers (I) the power reflectance, $R$, which is solely a function of the refractive index and (II) the optical thickness, $n_s d_s$, contained in the parameter $m$. For the investigated silicon wafers with comparatively low finesse (reflectance $R = 30\%$ only), the refractive index shows an error in the range of 1% while the optical thickness is obviously orders of magnitude more precise as we will prove next. As the geometrical thickness is calculated from the optical thickness and the refractive index, also the precision of the geometrical thickness is in the 1% range.

**Thickness resolution limits**. The ultimate optical thickness resolution is limited by the statistical fluctuations of the setup. The largest contribution to these is the frequency stability of the lasers, composed of the laser linewidth and any drifts occurring during the 24 s measurement time. While random fluctuations during the measurement duration hardly matter as the method averages over many frequency points, linear drifts cause errors in the phase evaluation. We estimate the stability to $\delta f_{THz} = 209 \pm 50$ kHz determined by terahertz spectral analysis and estimated from the error from the dependence on $\Delta(nd)$, shown in the supplemental material. For $m \sim n_s d_s \ll \Delta(nd) \sim m_0$ as used in most homodyne photomixing spectrometers, the dominant noise source originates from the second term in Eq. (10), causing a phase error of

$$\vartheta_{rms} = \frac{2\pi\Delta(nd)}{c_0}\delta f_{THz}, \qquad (11)$$

where $\Delta(nd)$ is the path length difference of the Mach–Zehnder arms and $\frac{2\pi\Delta(nd)}{c_0} = m + m_0$ in Eq. (10). The minimally detectable slope of a whole frequency sweep is

given by $\frac{\vartheta_{rms}}{f_{scan}}$, where $f_{scan}$ is the scanning range. With Eq. (11), the rms error of the optical thickness calculates to

$$\Delta(nd)_{rms} = \frac{c_0 \vartheta_{rms}}{2\pi f_{scan}} = \Delta(nd)\frac{\delta f_{THz}}{f_{scan}} \qquad (12)$$

The thickness error, $\Delta h$, and thus the resolution of a structure with refractive index $n_{obj}$ is thus given by

$$\Delta h = \frac{\Delta(nd)}{n_{obj}} \cdot \frac{\delta f_{THz}}{f_{scan}} \qquad (13)$$

For the 10.7 μm deep structure etched into silicon as shown in Fig. 2 a total optical path length difference $\Delta(nd) = 55$ cm and a scan window size of $f_{scan} = 200$ GHz was used. The geometrical thickness error according to Eq. (13) calculates to 183 nm, assuming the literature value of $n_{obj} = 3.416$. The measured rms precision was slightly higher with a value of 289 nm. For Fig. 3, path length differences of $\Delta(nd) = 7.25$ cm (240 nm Siemens star, $n_{obj} \sim 2.04$) and $\Delta(nd) = 2.4$ cm (49 nm Siemens star, $n_{obj} \sim 3.1$) were used, yielding a physical (optical) thickness error limit of 37 nm (75 nm) and 8 nm (25 nm) according to Eq. (13), respectively. A detailed examination of the thickness error for a variety of optical path length differences $\Delta(nd)$ is summarized in the supplemental material. However, imperfections in the setup, such as standing waves and other non-idealities severely increase the measurement error.

We developed two strategies to remove the deteriorating effect of these. First, we subtract the phase obtained from a reference measurement of the empty setup, $\Delta\vartheta_{ref}(f_{THz}) = \left(\frac{2\pi(n_s-1)d_s}{c_0} + m_0\right)f_{THz}$, prior to fitting with Eq. (10). The reference phase contains information about most imperfections and standing waves within the used optics. Only standing waves introduced or altered by the presence of the sample remain. It turned out to be sufficient to record only two reference spectra, one prior the first pixel and one after the last pixel. The linear part of Eq. (10) with subtracted reference measurement then becomes $\varphi_{lin} = k_0(n_s - 1)d_s$. As noise and laser fluctuations in the reference measurement and sample measurement are linearly independent, the fundamental noise limit from Eq. (13) increases by a factor of $\sqrt{2}$. Extraction of the geometric thickness from the slope only requires previous knowledge of the refractive index.

Second, the Fabry–Pérot term (1st term) of Eq. (10) yields much more precise fits, as long as the visibility of the Fabry–Pérot fringes is sufficient (a detailed discussion is given in the supplemental document). The main difference to fitting just the linear component of Eq. (10) is that the Fabry–Pérot term is independent of $m_0$ and hence $\Delta(nd)$. It solely depends on the substrate and object's (Si, SiC, SiN) optical thickness, $n_s d_s$, and $n_{obj}d_{obj}$, respectively. The minimum detectable height is given by

$$\Delta h_{FP} \approx \frac{1}{\eta_{fit}} \cdot \frac{(n_s d_s + n_{obj}d_{obj})}{n_{obj}} \cdot \frac{\delta f_{THz}}{f_{scan}} \approx \frac{1}{\eta_{fit}} \cdot \frac{n_s d_s}{n_{obj}} \cdot \frac{\delta f_{THz}}{f_{scan}} \qquad (14)$$

where $\eta_{fit} \leq 1$ considers the fitting accuracy that depends on the total noise caused by $\Delta(nd)$ in Eq. (13) and the sample finesse. Still, the structure added by the Fabry–Pérot resonances is, in our case, much larger than the amplitude of undesired standing waves in the setup.

With these two strategies, the measured rms error was improved to be only about a factor of 2–4 larger than predicted by Eq. (13). It is remarkable that the thickness error is proportional to the laser stability, $\delta f_{THz}$, which can be reduced to the Hz-level when using comb-based systems[37].

For samples with high finesse, the term $\eta_{fit}$ may become independent of $\Delta(nd)$, enabling to reach a resolution limit below that of Eq. (13). Particularly for materials with high reflectance, the optical thickness obtained from the fit to the FP phase (parameter $m$ in the first term in Eq. (10)), or, alternatively, the FP amplitude, yields a much higher accuracy than the fit to the slope (second term in Eq. (10)) due to the structure of the resonances. For the concrete examples of the 240 nm SiN ($n_{obj} = 2.04$) and the 49 nm SiC ($n_{obj} \sim 3.1$) Siemens star on a 525 μm thick silicon wafer ($n_s d_s = 17.93$ mm) the theoretically minimum detectable height difference (assuming $\eta_{fit} \sim 1$) is 0.9 nm for SiN and 0.6 nm for SiC. This is about 50 times lower than the experimentally found rms error of 31 nm at $\Delta(nd) = 2.4$ cm.

## Data availability
The raw data supporting this study have been deposited in the "TUdatalib" database and are available at https://doi.org/10.48328/tudatalib-664.

## Code availability
All codes written for and used in this study are available from the corresponding authors upon request.

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

## Acknowledgements

This research is financially supported by the European Research Council (ERC) through the Starting Grant "Pho-T-Lyze", grant agreement number 713780.

## Author contributions

S.P. conceived and supervised the experiment, M.M.A., A.k.M., A.F.O. assisted in setting up the experiment, A.I.R. set up the experiment and developed the Hilbert transformation-based data evaluation algorithm with help of M.M.A. A.I.R. and A.k.M. performed the measurements. S.P. wrote the manuscript with the aid of the other coauthors.

## Funding

## Competing interests

The authors declare no competing interests.
