## [Peer Review File · Nature Communications]

REVIEWER COMMENTS

Reviewer #1 (Remarks to the Author):

With the topic of terahertz thickness measurement, the authors address an area of research which has attracted great attention in recent years. This has resulted in a large number of publications, which also deal intensively with signal processing and comprehensively address the physical measurement conditions. Consideration of multiple reflections and, correspondingly, interference between individual interfaces is common. Therefore, the highlighted aspects of the relevant research work in the underlying manuscript do not contain any significant novelties and I cannot recommend publication of the manuscript in the given journal. Terahertz systems for layer thickness determination are used today in automotive production and in the field of plastics extrusion. Typical standard deviations of less than 100 nm are achieved. Taking into account the dielectric properties of the sample, these values are in a similar range to those obtained in the underlying research work. In the applications mentioned, single layer thicknesses (often a few μm thick) are determined within multilayer systems in reflection geometry. Although the present work is based on a different measurement geometry with a transmission arrangement, there are also a number of relevant publications that address this issue, for example:

<https://doi.org/10.1364/OE.22.000972>.

The large number of publications in the field of terahertz thickness measurement shows that, in addition to the measurement geometries, other measurement modalities such as the signal shape, bandwidth and frequency resolution also play a decisive role in defining the application possibilities and limitations. Therefore, no aspects are addressed in this manuscript that cannot be derived from the current state of research. However, the addressed measurement example represents a very specific but quite interesting application possibility of the technology. A clear differentiation from previous publications in the research area will probably provide the space for a publication in a suitable journal for this specific use case.

Reviewer #2 (Remarks to the Author):

The paper demonstrates high resolution thickness measurements using a simple THz homodyne system. The achieved thickness resolution is three orders of magnitude smaller than the wavelength, which is an impressive achievement. The described method is simple and can be implemented using other homodyne systems. The paper is meticulously written and addresses relevant metrological aspects. I have only one query: in Fig 3 images a and e appear to be the same. Please check.

The paper should be accepted for publication.

Reviewer #3 (Remarks to the Author):

The work presents a technique to enhance depth resolution in terahertz homodyne detection. The authors implement the Hilbert transform to extract the depth from the fluctuating amplitude profile. I believe the work is interesting and worthwhile for a number of terahertz applications. I have some comments to improve the clarity of this paper.

In the abstract, the authors mentioned that the depth precision surpasses the state-of-the-art by 1-2 orders of magnitude. Is the state-of-the-art being referred to a nearfield or far-field system?

The authors provide useful information in the method section later. Those points should be referred to from the main content for clarity. For example, the authors apply a fitting model in Fig.

1(c) but I was not sure which model the authors used. Same for the depth precision. It is discussed much later as a function of the frequency scan range. This should be briefly discussed in the main part too.

It is interesting that the Hilbert transform can resolve the depth at a fraction of the wavelength. Is there any physical explanation to this? In the current form, this implication is not discussed at all.

The estimated index of silicon is said to be 3.47, but earlier the authors mentioned 3.416. Which one is correct?

How did the author retrieve the wafer baseline or wrapped background in Fig. 3e?

All samples demonstrated are low loss. Will the loss cause a problem in the estimation?

Line 163 on p5, the authors mention low tuning speed. Please be specific. What is this tuning speed?

RESPONSE TO REVIEWERS

Reviewer 1

R1.1. With the topic of terahertz thickness measurement, the authors address an area of research which has attracted great attention in recent years. This has resulted in a large number of publications, which also deal intensively with signal processing and comprehensively address the physical measurement conditions. Consideration of multiple reflections and, correspondingly, interference between individual interfaces is common. Therefore, the highlighted aspects of the relevant research work in the underlying manuscript do not contain any significant novelties and I cannot recommend publication of the manuscript in the given journal. Terahertz systems for layer thickness determination are used today in automotive production and in the field of plastics extrusion. Typical standard deviations of less than 100 nm are achieved. Taking into account the dielectric properties of the sample, these values are in a similar range to those obtained in the underlying research work. In the applications mentioned, single layer thicknesses (often a few μm thick) are determined within multilayer systems in reflection geometry. Although the present work is based on a different measurement geometry with a transmission arrangement, there are also a number of relevant publications that address this issue, for example: <https://doi.org/10.1364/OE.22.000972>.

The large number of publications in the field of terahertz thickness measurement shows that, in addition to the measurement geometries, other measurement modalities such as the signal shape, bandwidth and frequency resolution also play a decisive role in defining the application possibilities and limitations. Therefore, no aspects are addressed in this manuscript that cannot be derived from the current state of research. However, the addressed measurement example represents a very specific but quite interesting application possibility of the technology. A clear differentiation from previous publications in the research area will probably provide the space for a publication in a suitable journal for this specific use case.

- We agree with the reviewer that thickness measurement is a vivid field of research. We honor the results achieved with time domain systems. However, we did a very thorough literature research and did not find the 100 nm deviations mentioned by the reviewer. The best value we found was around 300 nm, estimated from the images in papers ["Highly accurate thickness measurement of multi-layered automotive paints using terahertz technology," doi: <https://doi.org/10.1063/1.4955407>, "Fully Automated Terahertz Layer Thickness Measurement System," doi: [10.1109/IRMMW-THz46771.2020.9370502](https://doi.org/10.1109/IRMMW-THz46771.2020.9370502)]. Also in the review suggested by the reviewer, the best value mentioned in table 1 is 0.19 μm (i.e. 6 times more than reported in our measurement). This value, however, is a theoretical limit for the resolution based on the refractive indices from refs 16 and 18 of that paper. Further, the measurements in these references are done with plane wafers with a larger THz spot, possibly a collimated beam, where the THz beam averages over a larger area, reducing influences from any surface imperfections. These are no imaging applications, even the thickness of the investigated samples was measured by other means, maybe mechanical, neglecting the measurement error thereof (possibly in several microns!). In concurrence with the reviewer, we note that imaging applications are different from thickness determination, as in the former case, one has to consider the local variations such as surface roughness and imperfections, which adversely affect the overall standard deviation of the measured area, which in our case is as big as $25 \times 25 \text{ mm}^2$. The visible light photographs of our structures indeed show significant surface roughness and we believe that part of our error bar is due to this roughness, not due to the measurement technique. The other two examples in table 1 of the reference mentioned by the reviewer (that are by far more recent than refs. 16 and 18) show standard deviations of the order of 0.67 – 1.37 μm . If the reviewer can send us a

manuscript that shows the 100 nm thickness error, we would be happy to include it as a reference and we

would then rephrase that the system is at least 1 order of magnitude better than any so far reported CW system (with 15 times larger employed bandwidth) and at the same order as the best time domain systems, despite more than 20 times less bandwidth. In order to clarify this point, we have modified the final statement at the end of the introductory part:

“In this paper, we demonstrate a Terahertz far field system working at imaging wavelengths between 0.375 mm and 0.5 mm with the ability to visualize surface structures with heights of only 49 nm (~ /10000). The root mean square error (i.e. the precision) is as small as 31 nm (~ /15000) on a silicon sample that is as thick as 0.5 mm, despite the employed bandwidth of 0.2 THz is at least 15 times smaller than that of the THz systems with the best thickness resolution to date.” In the description of the experiment, we further clarified (line 108/109): *“..., i.e. using a small fraction of the frequency range usually employed in thickness measurements”*.

So we still believe that our claim of showing at least one order of magnitude higher resolution than the state of the art is still correct, at least for continuous-wave measurements.

We reduced the statement in the abstract to *“This precision surpasses the far field state-of-the-art by 1 order of magnitude.”*

- We believe that the reviewer mainly refers to pulsed TDS measurements (automotive paints and plastic tubing investigations, to our knowledge, are done with TDS systems). These, however, feature a bandwidth typically beyond 4 THz (single shot bandwidth, see also our paper: <https://doi.org/10.1364/OL.388870>). The spatial resolution scales roughly with pulse duration and therefore inversely with frequency coverage (assuming diffraction-limited pulses). Assuming that 100 nm is a typical standard deviation of pulsed systems, the 20 times smaller bandwidth of the CW system would imply a resolution not better than 2 m.
- If a similar bandwidth of > 4 THz as in typical TDS systems was used in our proposed continuous-wave setup (CW systems yet offer indeed 4 THz bandwidth, see L. Liebermeister et al, Nature Communications **12**, 1071 (2021); however a 4 THz scan would take way too long at the moment), the high bandwidth translates into a *theoretical optical thickness error limit* of our proposed system as low as 1.25 nm according to Eq. M13, which is more than two orders of magnitude lower than the current state-of-the-art systems. We have added the following discussion of this point in the revised manuscript (lines 196-202):

“Measurement precision can be further improved by scanning over a larger bandwidth. As shown in eq. M13 and M14, the measurement precision h is inversely proportional to the scanning bandwidth ($\Delta\nu$). Additionally, scanning over a larger frequency range will increase the fitting efficiency (χ^2) as higher numbers of Fabry-Pérot fringes will ensure a better fit. Thus, by simply increasing the scanning bandwidth to 2-4 THz, which is usually the case for the state-of-the-art thickness measurement systems, the measurement precision can be improved to sub-nanometer levels, given that other measurement imperfections do not limit the precision.”

The well-cited paper mentioned a critical thickness (h_c) of a thin film (irrespective of whether it is on a substrate or free standing) in equation (13), which is the minimum detectable film thickness for phase-based estimates. The authors of the referred paper also state that for accurate thickness measurements in transmission geometry, the thickness of the thin-film should be at least 10 times higher than this critical thickness. We used the same equation for our data and the corresponding critical thickness values are plotted in Fig. 1 as a function of frequency. It is evident from the plot that, for the material combination used in our work, the average minimum detectable thickness should be ≈ 1.24 m, however, we have comprehensively determined the thickness of a 50 nm SiC ($n = 3.1$) Siemens star deposited on a highly resistive silicon ($n = 3.416$) substrate with a ± 30 nm error margin. This is already a factor of 20 less than the predicted minimum measurable thickness in the referred paper and two orders of magnitude less than the recommended measurable thickness. Surprisingly, the paper presents no comparison between the actual and

measured thickness of any photoresist film or other film-substrate combination to substantiate their claims.

Fig. 1 For the calculations we used $n = 1$, as is the case in Table 1 of the referred paper, $m = 3.1$, and $\theta = 3.416$. We calculated $t_{\text{c,p}}$ from a series of $N = 25$ phase scans over a frequency range of 0.6 – 0.78 THz

- We completely agree with the reviewer that the thickness measurement of thin films depends on the measurement setup and mentioned measurement modalities. That is why we have already incorporated these factors in our calculations presented in equations (M13) and (M14). We have acknowledged these setup-related limitations and circumvented those to achieve as low as 20 nm measurement standard deviation. Extraordinary laser stabilization or using dry air can further improve these reported values. As an example, Fig. S1 of the supplemental material shows that just by changing the optical path length difference in the homodyne setup, the estimation error can be significantly reduced.
- We believe the novelty of this work lies in its simplicity, wherein an off-the-shelf commercial homodyne Terahertz system can be used to measure thickness up to five orders of magnitude smaller than the operating wavelength, without employing any extraordinary stabilization technique or any artificial or neural network-based data evaluation with a large requirement on computation power. We have comprehensively considered and incorporated all system related errors in our calculations done in the methods section and proposed ways to minimize them. We further believe that with advanced data processing techniques, such as multi-pixel image processing or neural networks that are, to our knowledge, used in the highest thickness resolution TDS systems we can further reduce our standard deviation. This, however, is part of future work. We therefore added at the end of the conclusions section: "Advanced data extraction techniques, such as neural networks that are yet implemented in many state-of-the-art Terahertz thickness measurement systems may further improve the precision."
- The only limitation we could not address in this work is the comparatively high measurement time required due to pixel-wise scanning technique, as we do not currently possess any homodyne receiver with multiple pixels. However, this issue can be solved by using a THz camera that is transformed into a homodyne detector with the use of a local oscillator, i.e. a photomixer driven by the same signal used for sample illumination. We have discussed this issue in the supplemental information.

Reviewer 2

R2.1. I have only one query: in Fig 3 images a and e appear to be the same. Please check.

- Images (a) and (e) in Fig. 3 are different images. If inspected closely, traces of the 250 nm Siemens star is slightly visible in (a), which is the raw measurement data, initially obtained from the measurement setup. We later subtract the warping plane, presented in (e), from the raw data to obtain the clear image of the Siemens star shown in (b). The reason why they look so similar is the fact that the height of the Siemens star is 250 nm whereas the curvature/non-flatness of the substrate is larger than $\approx 5 \mu\text{m}$ over the measured area, i.e. 20 times larger than the structure height.

Reviewer 3

R3.1. In the abstract, the authors mentioned that the depth precision surpasses the state-of-the-art by 1-2 orders of magnitude. Is the state-of-the-art being referred to a nearfield or far-field system?

- We refer to the far-field systems in the abstract. We have rephrased the sentence in abstract (line 15).

R3.2. The authors provide useful information in the method section later. Those points should be referred to from the main content for clarity. For example, the authors apply a fitting model in Fig. 1(c) but I was not sure which model the authors used. Same for the depth precision. It is discussed much later as a function of the frequency scan range. This should be briefly discussed in the main part too.

- Thank you for the remark. In order to make the paper better understandable, we have added explanatory paragraphs in the revised manuscript in lines 98-100, 111-114, 140-143. Further, we have added references in the main text to the equations in the methods section.

R3.3 It is interesting that the Hilbert transform can resolve the depth at a fraction of the wavelength. Is there any physical explanation to this? In the current form, this implication is not discussed at all.

- Indeed, this phenomenon is discussed very well by Vogt et al, ref. 30. An abbreviated discussion is actually in the main text in the first paragraph on page 3. In fact, the Hilbert transformation as such does not impact the depth resolution, at least from a theoretical perspective. It is simply a very clever trick (first applied by Vogt et al, ref 30), to get the phase and amplitude information from the homodyne fringes without the need of fitting a sine wave to very high precision. This mathematical trick severely improves the data quality of the phase and amplitude extraction, even for fractions of a homodyne fringe. The physical reason why it is possible is that the periodicity of the homodyne fringes change, also on the scale of a fraction of the sine wave, if the optical path length changes. In our case, the condition is actually relaxed: imagine we have a change of the optical path length by the presence of the sample. Then the path length difference between receiver and source/THz arm change, so does the periodicity of the homodyne fringes as discussed in the text following equation M2. Acquiring many fringes strongly reduces the measurement error. The reason why we limited ourselves to a scan range of 200 GHz (4001 data points) was simply measurement time. Our system can theoretically go up to 2.75 THz but this would have been impractical. In fact, we do not (yet) evaluate the fringe frequency directly, we rather evaluate

its phase. But the information in there follows the same rules: the slope of the phase changes and the Fabry-Pérot pattern becomes visible which further enhances the accuracy drastically.

R3.4. The estimated index of silicon is said to be 3.47, but earlier the authors mentioned 3.416. Which one is correct?

- Disentangling the refractive index and the physical thickness increases the error for each. The calculated refractive index from the measured sample shown in Fig. 1(b) is 3.47, has around 2% measurement error as simultaneous determination of refractive index and thickness was carried out in this case, where the optical path length difference in the setup, $A(nd) = 55$ cm. This measurement error can be drastically reduced if a smaller value of $A(nd)$ is used or thickness of the sample is considered to be a constant for refractive index calculation, as observed in Fig. S1 of the supplemental file. The accuracy on $A(nd)$ is much higher than the 2% mentioned here, enabling us to visualize the 50 nm Siemens star. We have rephrased the sentence to make it clear (lines 120122).

R3.5. How did the author retrieve the wafer baseline or wrapped background in Fig. 3e?

- The whole measurement data was fitted to a fifth order 2-D polynomial to estimate the slow trend of changing substrate wafer thickness, say $A(a)$, which for our samples is ≈ 3 m. This is the wrapped background shown in Fig. 3e. Since the height difference due to the deposited layer for 50, 250 and 350 nm samples is much smaller and localized than $A(a)$, the 2-D polynomial fit successfully mimics the wafer surface. However, if the height difference of the deposited/etched structures becomes larger than $A(a)$, as in the case of 10 m measurement, the calculated wrapped background actually over-corrects the image, as shown in Fig. S9 in the supplemental document. So, the estimation of a 10 m deep structure necessitates not using background subtraction.

R3.6. All samples demonstrated are low loss. Will the loss cause a problem in the estimation?

- We expect the measurement error to deteriorate for lossy materials, and consequently minimum detectable thickness will increase. The reason is that the finesse of the Fabry-Pérot resonances will go down, so the fitting algorithm will get less accurate. We would still be able to see the average slope from where we could also calculate the optical thickness, however, with reduced accuracy. To verify this, we measured a 1.98 ± 0.02 mm thick (measured with digital calipers) PVC sample ($n = 1.64$, ≈ 11 cm⁻¹ at 0.7 THz) with $A(nd) = 4.1$ cm over the same bandwidth and evaluated the mean sample thickness to be 1.979 mm without any change in the evaluation algorithm. The measurement precision is around 5 m, i.e. 0.25% of the actual sample thickness. Additionally, we can also look at the homodyne fringe period of our measurement, which is proportional to $A(nd)/\omega$, to measure the change in optical path length difference caused by absorptive materials and estimate the material thickness. The measurement precision will also be lower in this case, given we receive enough signal to resolve the homodyne fringes appropriately.

R3.7. Line 163 on p5, the authors mention low tuning speed. Please be specific. What is this tuning speed?

- The *tuning speed* corresponds to the tuning speed of the two temperature controlled DFB lasers used in this experiment plus the waiting time at each point caused by the used lock-in time constant of 3 ms. One sweep of 200 GHz therefore takes about 12 s with our laser system. We further opted to wait 30 seconds for stabilization before taking the next sweep as we use a saw tooth modulation. Triangular modulation would to a large extent remove the necessity of the waiting time.

Upon calibration, a mapping is done between the temperature of the lasers and corresponding terahertz frequency generated due the mixing of these two lasers. This data is stored in a look-up table in the system software. The rate of temperature change during calibration, however, was slower than what we have used during measurements. So there is an additional frequency offset introduced due to the time-lag between set laser-temperature and actual laser-temperature. However, we noticed that the time lag is quite constant such that the time lag only produces an overall offset of the phase (likewise, an offset of the THz frequency of the order of 2 GHz or so). As we only evaluate the slope of the phase and the modulation introduced by the Fabry-Pérot oscillations, the measurement technique is immune against constant offsets. We have rephrased lines 169-173 to make it clearer.

REVIEWERS' COMMENTS

Reviewer #1 (Remarks to the Author):

I appreciate the thorough and detailed elaboration of the authors and must acknowledge that the relevant reference quantities are often only indirectly addressed in the literature. This is probably primarily due to the fact that in applications of layer thickness measurement by means of terahertz radiation, there is often only one-sided access to the measurement sample or single layer thicknesses of multilayer systems without a priori information (such as the exact knowledge of the Si wafer in the present manuscript) have to be determined and therefore reflection measurements are required. Typically, the main focus is on resolution, i.e., the possible discrimination of two consecutive interfaces. The transmission method presented by the authors in conjunction with the simplified assumption of a single layer (Si substrate and applied structure) is therefore more comparable with the standard deviation of the measurement system in corresponding reflection measurement setups with regard to the structure thickness. This is unfortunately rarely mentioned in the literature, but can be found, for example, in the more recent publication by D. Molter et. al. <https://doi.org/10.1063/5.0037395>, which addresses, among other things, TDS systems with much wider bandwidth than the measurement system in the present work. The authors of the present work also explicitly refer to the much smaller measurement bandwidth in their work compared to corresponding TDS systems and address the frequency accuracy of their system. Besides the wavelength and the SNR of the measurement signal, essential factors that influence the measurement accuracy and the detectable structure thickness, which, however, cannot be considered separately. For example, B. Hils et. al. <https://doi.org/10.1364/OE.16.011289> use a quartz-stabilized CW source at a fixed frequency of 600 GHz and thus achieve accuracies of 500 nm (optical path length) in terahertz surface profilometry. Pohl et. al. <https://doi.org/10.1109/TMTT.2011.2180398> gives standard deviations of 360 nm for range measurements using FMCW systems (again, this relates to optical path lengths). The methods of all the above references allow the visualization of comparable nanoscale structures of the same order of magnitude (TDS:40nm, CW:500nm, FMCW:360nm optical path length)! I therefore continue to argue that the novelty of the present work lies rather in the interesting but very specific application than in the results obtained and recommend to publish the work in another more suitable journal. In any case, the discussion shows that a comprehensible elaboration of the correlations and the reference values is missing in the literature. Should the authors be interested in a corresponding review, a closer look at the work on material characterization in the microwave range, such as by D.K. Ghodgaonkar et. al. <https://doi.org/10.1109/19.32194>, is also recommended.

Two more minor comments:

The authors describe that the made assumption of identical material parameters of substrate and structure is subject to error and present the expected error magnitudes in the supplementary material. It should be noted that a priori information regarding the substrate material (Si wafer) is also mandatory for this consideration. Schreiner et. al. <https://doi.org/10.3390/s19183910> describe the mutual interference of several interfaces and the resulting difficulties in layer thickness determination.

The authors refer to the use of neural networks to determine layer thicknesses with highest resolution. This is certainly the subject of current research. In application, genetic algorithms, e.g. Zwick et. al. <https://doi.org/10.1109/22.993422> or model-based approaches as in the previously mentioned reference are widely used until today.

Reviewer #3 (Remarks to the Author):

The authors have addressed all my comments satisfactorily.

Dear Reviewers,

Thank you very much for the positive response and we thank you for your feedback that helps us to further improve the manuscript. We have tried to address the reviewers' comments to the best of our ability in this letter. Please find a list of corrections and improvement below.

Best regards,
Amlan kusum Mukherjee

Reviewer #1

I appreciate the thorough and detailed elaboration of the authors and must acknowledge that the relevant reference quantities are often only indirectly addressed in the literature. This is probably primarily due to the fact that in applications of layer thickness measurement by means of terahertz radiation, there is often only one-sided access to the measurement sample or single layer thicknesses of multilayer systems without apriori information (such as the exact knowledge of the Si wafer in the present manuscript) have to be determined and therefore reflection measurements are required. Typically, the main focus is on resolution, i.e., the possible discrimination of two consecutive interfaces. The transmission method presented by the authors in conjunction with the simplified assumption of a single layer (Si substrate and applied structure) is therefore more comparable with the standard deviation of the measurement system in corresponding reflection measurement setups with regard to the structure thickness. This is unfortunately rarely mentioned in the literature, but can be found, for example, in the more recent publication by D. Molter et. al. <https://doi.org/10.1063/5.0037395>, which addresses, among other things, TDS systems with much wider bandwidth than the measurement system in the present work. The authors of the present work also explicitly refer to the much smaller measurement bandwidth in their work compared to corresponding TDS systems and address the frequency accuracy of their system. Besides the wavelength and the SNR of the measurement signal, essential factors that influence the measurement accuracy and the detectable structure thickness, which, however, cannot be considered separately. For example, B. Hils et. al. <https://doi.org/10.1364/OE.16.011289> use a quartz-stabilized CW source at a fixed frequency of 600 GHz and thus achieve accuracies of 500 nm (optical path length) in terahertz surface profilometry. Pohl et. al. <https://doi.org/10.1109/TMTT.2011.2180398> gives standard deviations of 360 nm for range measurements using FMCW systems (again, this relates to optical path lengths). The methods of all the above references allow the visualization of comparable nanoscale structures of the same order of magnitude (TDS:40nm, CW:500nm, FMCW:360nm optical path length)! I therefore continue to argue that the novelty of the present work lies rather in the interesting but very specific application than in the results obtained and recommend to publish the work in another more suitable journal. In any case, the discussion shows that a comprehensible elaboration of the correlations and the reference values is missing in the literature. Should the authors be interested in a corresponding review, a closer look at the work on material characterization in the microwave range, such as by D.K. Ghodgaonkar et. al. <https://doi.org/10.1109/19.32194>, is also recommended.

- We acknowledge that TDS systems can achieve similar thickness resolution, however, the measurement bandwidths for these systems are at least 10 times higher than the measurement bandwidth employed in our setup. Given access to such huge bandwidths, equation M13 predicts an optical thickness error value less than 2.5 nanometers, i.e. an order of magnitude better than the TDS measurement values presented by D. Molter et al. (<https://doi.org/10.1063/5.0037395>). This article was, however, published after our initial submission and hence, we did not have access

to this information while preparing this manuscript. We have removed the statement “This precision surpasses the far field state-of-the-art by about 1 order of magnitude.” from the abstract (line 14) of our latest revision. We also agree with the reviewer, that a reflection setup is yet to be demonstrated, mathematically, there is, however, no severe difference. The main difference is that our substrate is part of the measurement. The CW minimum thickness values/standard deviations obtained in the mentioned references with 500 nm and 360 nm are still a factor of 2-4 larger. Additional stabilizing of the laser systems, as done by B. Hils et al. (<https://doi.org/10.1364/OE.16.011289>), will further improve the achieved resolution, however this also increases the complexity and cost of the measurement system. Phase locking, as employed by Pohl et al. (<https://doi.org/10.1109/TMTT.2011.2180398>), is already inherent to our setup as same DFB-Lasers are used for both transmitter and receivers, as we have mentioned in line 322. We have also acknowledged this fact in our conclusion that the measured resolution can be improved upon by using various laser stabilization techniques, noise reduction and machine learning algorithms. We have added D. Molter et al. as benchmark for thickness determination using TDS systems in lines 217-218.

Two more minor comments:

The authors describe that the made assumption of identical material parameters of substrate and structure is subject to error and present the expected error magnitudes in the supplementary material. It should be noted that apriori information regarding the substrate material (Si wafer) is also mandatory for this consideration. Schreiner et. al. <https://doi.org/10.3390/s19183910> describe the mutual interference of several interfaces and the resulting difficulties in layer thickness determination.

- We indeed did not look into multi-layer structures so far and recognize the important work by Mrs. Schreiner as essential for multi-layer stack measurements. Considering the substrate, we agree with the reviewer that some apriori information about the host wafer, especially its refractive index, is essential to determine the appropriate frequency range of measurement and fitting bounds for data evaluation. However these information can be extracted from the measurements. Calculating optical thickness is less erroneous than calculating its geometrical thickness and refractive index (c.f. Fig. S1). Thus, in the employed data evaluation algorithm, the optical thickness of the substrate is first calculated to a precision of ≈ 70 nm with a $\Delta(nd) \approx 2$ cm and is, subsequently, divided by mean value of the measured refractive index of the substrate (refractive index error = 0.004 with $\Delta(nd) \approx 2$ cm) to get material parameters of the host silicon wafer. This serves as the apriori material information, which is subsequently used in the fitting algorithm. We have added lines 204-209 to address this issue.

The authors refer to the use of neural networks to determine layer thicknesses with highest resolution. This is certainly the subject of current research. In application, genetic algorithms, e.g. Zwick et. al. <https://doi.org/10.1109/22.993422> or model-based approaches as in the previously mentioned reference are widely used until today.

- Thank you for pointing out the literature on data evaluation techniques using neural networks. We have added the literature as reference in our final submission in line 221.

Reviewer #3

The authors have addressed all my comments satisfactorily.

- We thank the reviewer for accepting our changes and modifications.